# Flavopereirine Inhibits Autophagy via the AKT/p38 MAPK Signaling Pathway in MDA-MB-231 Cells

**DOI:** 10.3390/ijms21155362

**Published:** 2020-07-28

**Authors:** Ming-Shan Chen, Hsuan-Te Yeh, Yi-Zhen Li, Wen-Chun Lin, Ying-Ray Lee, Ya-Shih Tseng, Shew-Meei Sheu

**Affiliations:** 1Department of Anesthesiology, Ditmanson Medical Foundation Chia-Yi Christian Hospital, Chia-Yi 60002, Taiwan; 06590@cych.org.tw (M.-S.C.); 07438@cych.org.tw (H.-T.Y.); 2Department of Biotechnology, Asia University, Taichung 41354, Taiwan; 3Department of Medical Research, Ditmanson Medical Foundation Chia-Yi Christian Hospital, Chia-Yi 60002, Taiwan; 10862@cych.org.tw (Y.-Z.L.); 13320@cych.org.tw (W.-C.L.); 06841@cych.org.tw (Y.-R.L.); 4Department of Medical Laboratory Science and Biotechnology, Chung Hwa University of Medical Technology, Tainan 71703, Taiwan; shih415@mail.hwai.edu.tw

**Keywords:** flavopereirine, autophagy, breast cancer, AKT, p38 MAPK

## Abstract

Autophagy is a potential target for the treatment of triple negative breast cancer (TNBC). Because of a lack of targeted therapies for TNBC, it is vital to find optimal agents that avoid chemoresistance and metastasis. Flavopereirine has anti-proliferation ability in cancer cells, but whether it regulates autophagy in breast cancer cells remains unclear. A Premo™ Tandem Autophagy Sensor Kit was used to image the stage at which flavopereirine affects autophagy by confocal microscopy. A plasmid that constitutively expresses p-AKT and siRNA targeting p38 mitogen-activated protein kinase (MAPK) was used to confirm the related signaling pathways by Western blot. We found that flavopereirine induced microtubule-associated protein 1 light chain 3 (LC3)-II accumulation in a dose- and time-dependent manner in MDA-MB-231 cells. Confocal florescent images showed that flavopereirine blocked autophagosome fusion with lysosomes. Western blotting showed that flavopereirine directly suppressed p-AKT levels and mammalian target of rapamycin (mTOR) translation. Recovery of AKT phosphorylation decreased the level of p-p38 MAPK and LC3-II, but not mTOR. Moreover, flavopereirine-induced LC3-II accumulation was partially reduced in MDA-MB-231 cells that were transfected with p38 MAPK siRNA. Overall, flavopereirine blocked autophagy via LC3-II accumulation in autophagosomes, which was mediated by the AKT/p38 MAPK signaling pathway.

## 1. Introduction

Autophagy is a cellular self-digestion pathway that removes unnecessary or dysfunctional cellular components via sequestration in double membrane vesicles, which subsequently fuse with lysosomes for acidic degradation. Basal autophagy levels in cells serve to maintain homeostasis during nutrient deprivation, including clearance of damaged organelles and nutrient recycling. In addition to the physiological roles of autophagy, autophagic dysfunction has been identified in the pathogenesis of diverse diseases, including cancer, neurodegeneration, microbial infection, and aging [1]. In cancer treatment, autophagy also plays a critical role in the survival of breast cancer cells and is considered a novel therapeutic target [2,3,4].

Breast cancer is the most frequently diagnosed cancer among women worldwide and the leading cause of cancer-related death in women [5]. Breast cancer is highly heterogeneous. Among the different subtypes, triple negative breast cancer (TNBC) is negative for hormonal (estrogen and progesterone) receptors and human epidermal growth factor receptor 2 (HER2) and thus lacks targeted therapy [6]. TNBC is rapidly progressive, with earlier onset of visceral metastases and poor prognosis [7,8]. Finding an optimal chemotherapy that leads to a longer metastasis-free period and overall survival for TNBC patients is urgently needed. Inducing autophagy was demonstrated to be a potential therapeutic strategy for TNBC treatment [9]. However, it is still unclear whether autophagy in TNBC has a protective or cytotoxic role [4].

Flavopereirine is a *β*-carboline alkaloid that can be extracted from the tree bark of several *Geissospermum* species [10] or chemically synthesized. Synthetic flavopereirine has been reported to selectively reduce human glioblastoma (U251) cell proliferation but only slightly inhibit normal astrocytes (CRL 1656) at a high concentration of 50 μg/mL [11]. Recent studies further demonstrated that flavopereirine significantly suppressed colorectal cancer cell growth in vitro and in vivo and induced apoptosis in breast cancer cells [12,13]. However, whether flavopereirine has an effect on autophagy in breast cancer remains unclear.

Recently, we demonstrated that flavopereirine induced TNBC (MDA-MB-231) cell cycle arrest and apoptosis through the AKT/p38 mitogen-activated protein kinase (MAPK)/extracellular regulated kinase (ERK)1/2 signaling pathway [13]. Because autophagy has become a potential therapeutic target for breast cancer, we further investigated whether flavopereirine has a role in autophagy regulation. In this study, we used flavopereirine-treated MDA-MB-231 cells to examine the mechanism by which flavopereirine is involved in autophagy.

## 2. Results

### 2.1. Flavopereirine Blocked Autophagic Flux

Autophagic flux is often examined through the turnover of the autophagy marker LC3 (LC3-II) by Western blotting. We tested the significance of flavopereirine in autophagy in MDA-MB-231 cells and found that flavopereirine induced the accumulation of LC3-II in a dose- and time-dependent manner (Figure 1). To confirm the localization of LC3-II, a Premo™ Tandem Autophagy Sensor Kit (Life Technologies, Carlsbad, CA, USA) was used to visualize autophagosome fusion with lysosomes to form autolysosomes by fluorescent protein signals (LC3-II fused with green fluorescent protein (GFP) and red fluorescent protein (RFP)). The structures of autophagosomes (neutral pH) were positive for both GFP (green) and RFP (red), which are shown as yellow/orange puncta. However, the pH-sensitive fluorescence of GFP was quenched by fusion with lysosomes, causing autolysosomes (acidic pH) to appear as red puncta. As shown in Figure 2, chloroquine (CQ), an autophagy inhibitor, blocked endogenous autophagic flux, leading to the accumulation of LC3-II in autophagosomes (yellow/orange). MDA-MB-231 cells treated with flavopereirine also accumulated autophagosomes, similar to the effect of CQ. The high concentration of flavopereirine (15 μM) increased the intensity of LC3-II puncta compared with that of the low dose (10 μM). Moreover, flavopereirine further blocked aloperine-induced autophagic flux, similar to the effect of CQ. These results indicated that flavopereirine arrests autophagic flux.

### 2.2. The Akt/p-p38 MAPK Pathway Was Targeted by Flavopereirine to Inhibit Autophagy

The Akt-mTOR pathway is a well-known pivotal negative signaling pathway that regulates autophagy. The kinase mTOR plays a core role in controlling autophagy initiation [14]. We found that the level of p-AKT decreased with exposure to higher concentrations of flavopereirine (10 and 15 μM) (Figure 3A). After MDA-MB-231 cells were transfected with a plasmid that constitutively expressed p-AKT, flavopereirine-induced LC3-II accumulation diminished (Figure 3B). Western blot imaging also showed that flavopereirine inhibited the basal expression levels of mTOR in MDA-MB-231 cells, leading to the suppression of p-mTOR. However, transfection with the p-AKT plasmid did not obviously rescue this suppression (Figure 3B,C).

The role of MAPK in flavopereirine-mediated autophagy was explored, and the flavopereirine-induced expression of p-p38 MAPK was lower in MDA-MB-231 cells transfected with the p-AKT plasmid than in cells transfected with the control plasmid (Figure 3C). After knocking down p38 MAPK with a specific siRNA, the flavopereirine-induced levels of total and phosphorylated p38 MAPK were reduced (Figure 3D). Flavopereirine-induced LC3-II accumulation also decreased when p38 MAPK was downregulated. These results indicated that p-AKT is an upstream inhibitor of p-p38 MAPK that mediates flavopereirine-induced LC3-II accumulation.

### 2.3. Flavopereirine Decreased Cell Viability More Than CQ

Treating MDA-MB-231 cells with 10 or 15 μM flavopereirine reduced cell viability in a dose-dependent manner, even at 24 and 48 h (Figure 4). Cell viability at these two concentrations was lower than in cells treated with 25 μM CQ. When flavopereirine and CQ were combined, cell viability was further reduced to almost complete cell death.

## 3. Discussion

There is a complicated interaction between autophagy and breast cancer progression, and the function of autophagy in breast cancer remains unclear [3,4]. Autophagy regulation is considered a possible treatment strategy for TNBC [9,15]. In the present study, we demonstrated that flavopereirine caused LC3-II accumulation in MDA-MB-231 cells in a dose- and time-dependent manner. Flavopereirine inhibited autophagic flux partially through the AKT/p38 MAPK pathway.

Treatment of TNBC with a combination of histone deacetylase inhibitors and ionizing radiation induces autophagic cell death [16]. The small molecule RL71 can also trigger excessive autophagic cell death in TNBC [9]. However, autophagy inhibition was more effective in TNBC than in luminal cell lines [15]. The four TNBC cell lines analyzed, including MDA-MB-231, were sensitive to cell death through autophagy inhibition by autophagy-related gene knockdown or CQ treatment, suggesting that autophagy inhibition might be a potential therapeutic strategy for TNBC. Our results provide a candidate that can inhibit basal and aloperine-induced autophagy leading to LC3-II accumulation in MDA-MB-231 cells (Figure 1 and Figure 2). In contrast to LC3-II accumulation, flavopereirine inhibited p62 in a dose-dependent manner (Figure 1). Previously, p62 was considered an autophagy indicator due to its ability to deliver polyubiquitinated proteins to autophagy degradation. Recent reports demonstrated that p62 participates in the regulation of diverse cellular signaling pathways [17], which may interfere with p62 expression or degradation independent of autophagy. These support that the p62 level alone is not a good indicator of autophagy. Besides CQ and hydroxychloroquine (HCQ) being the only clinically available drugs that inhibit autophagy for cancer treatment in clinical trials [18], flavopereirine presents the potential benefit of autophagy inhibition when applied to TNBC treatment.

AMP-activated protein kinase (AMPK) is the upstream energy receptor that is triggered by various stresses to activate autophagy [19]. Activation of AMPK requires specific phosphorylation of the α subunit at Thr172. AMPKα is phosphorylated at low levels in most of the investigated breast cancer cell lines (MCF-7, MDA-MB-468, MDA-MB-231, and MDA-MB-436), especially in MDA-MB-231 cells [20]. We also found that p-AMPKα expression was low in flavopereirine-treated MDA-MB-231 cells (data not shown), suggesting its minor role in flavopereirine-mediated autophagy inhibition. The activation of AKT/mTOR is crucial for the induction of autophagy [14]. Flavopereirine suppressed the translation of mTOR, subsequently resulting in decreased phosphorylation of mTOR (Figure 3B, C), which did not obviously change in MDA-MB-231 cells transfected with p-AKT plasmids (Figure 3C). We further demonstrated that AKT is upstream of p38 MAPK, which is mediated by flavopereirine to induce LC3-II accumulation (Figure 3C,D). These results indicated that flavopereirine partially inhibits autophagy through nonclassical AKT/p38 MAPK signaling.

The generation of cytotoxic autophagy may either lead independently to cell death or act as a precursor to apoptosis [3,4]. In our previous study, we used a pan-caspase inhibitor (z-VAD-fmk) to pretreat MDA-MB-231 cells, which significantly reduced flavopereirine-induced apoptosis from 77.2 to 27.5% at 48 h [13]. The proportion of cells that were positive for PI but not for annexin-V staining was 3.8%. These data suggested that other mechanisms could account for flavopereirine-mediated cytotoxicity other than caspase-dependent apoptosis. In this study, CQ, an autophagy inhibitor, decreased cell survival by more than 40% in MDA-MB-231 cells at 48 h (Figure 4). Treatment with flavopereirine inhibited endogenous autophagy (Figure 2) and resulted in lower cell survival than CQ treatment (Figure 4). MDA-MB-231 cells treated with a combination of flavopereirine and CQ showed further decreased cell survival compared to that of CQ treatment alone. These data suggested that the decrease in cell survival was due to flavopereirine-induced apoptosis and autophagy-related cell death.

In summary, flavopereirine inhibits endogenous autophagy partially through the AKT/p38 MAPK pathway, which is related to decreased cell survival.

## 4. Materials and Methods

### 4.1. Breast Cancer Cell Line and Culture Conditions

The MDA-MB-231 human breast cancer cell line was provided by Dr. Ying-Ray Lee. MDA-MB-231 cells were maintained in low glucose Dulbecco’s modified Eagle’s medium (Gibco-BRL, Carlsbad, CA, USA) supplemented with 8% fetal bovine serum and cultured in a humidified atmosphere with 5% CO_2_ at 37 °C.

### 4.2. Reagents and Antibodies

Flavopereirine perchlorate was obtained from ChromaDex, Inc. (Irvine, CA, USA). Aloperine (ab143290) was purchased from Abcam (Cambridge, MA, USA). A non-ATP-competitive MEK inhibitor, PD98059 (TargetMol, Boston, MA, USA), was used to suppress the phosphorylation of ERK. Antibodies targeting AKT (#4691), phospho-AKT (Ser473) (#4060), mammalian target of rapamycin (mTOR) (7C10) (#2983), phospho-mTOR (Ser2448) (#5536), p38 MAPK (#9212), phospho-p38 MAPK (Thr180/Tyr182) (#9211), p44/42 MAPK (ERK1/2) (#4695), and phospho-p44/42 MAPK (Thr202/Tyr204) (#4377) were purchased from Cell Signaling Technology, Inc. (Beverly, MA, USA). A polyclonal antibody to detect the autophagosomal marker protein microtubule-associated protein 1 light chain 3 (LC3) (PM 036) was obtained from Medical & Biological Laboratories Co. (Nagoya, Japan). An antibody against the p62 protein, also called sequestosome 1 (SQSTM1), (sc-28359) was purchased from Santa Cruz Biotechnology (Santa Cruz, CA, USA).

### 4.3. Imaging Autophagy with the RFP-GFP-LC3B Kit

MDA-MB-231 cells (1 × 10^4^ cells/well) were plated in a Nunc™ 177,437 Lab-Tek Chamber Slide System (Thermo Fisher Scientific, Rochester, NY, USA) and allowed to adhere overnight. The various stages of autophagy were monitored by the Premo™ Autophagy Tandem Sensor RFP-GFP LC3B Kit (Life Technologies, Carlsbad, CA, USA) according to the manufacturer’s instructions, which achieved efficient transduction using insect Baculovirus carrying the acid-sensitive LC3B-fluorescent protein chimera with a mammalian promoter. After the cells were treated overnight with 3 μL of BacMam reagents containing RFP-GFP-LC3B DNA, specified concentrations of drugs were subsequently added to the cell medium for 48 h. Aloperine (100 μM) was the positive control and induced autophagic flux, which was blocked by CQ. The media were removed, and live cell imaging solution containing Hoechst 33342 (1 µg/mL) was added and incubated for 20 min in the dark. The cells were then washed with 1× phosphate-buffered saline (PBS), covered in mounting medium (F4680) (Sigma-Aldrich, Inc. St. Louis, MO, USA), and imaged using a Zeiss laser scanning confocal microscope LSM800 (Carl Zeiss Microscopy GmbH, Jena, Germany).

### 4.4. Western Blotting

Cells were lysed with the M-PERTM mammalian protein extraction reagent (Thermo Fisher Scientific Inc., Rockford, IL, USA) containing a 0.1% protease inhibitor cocktail. Equal amounts of each sample (40 μg of protein) were loaded and separated on sodium dodecyl sulfate-polyacrylamide gel electrophoresis (SDS-PAGE) gels and then transferred to polyvinylidene fluoride (PVDF) membranes. Primary antibodies and horseradish peroxidase-conjugated secondary antibodies were used to target the specific proteins. The band signals were developed with Immobilon Western Chemiluminescent HRP Substrate (EMD Millipore Corporation, Billerica, MA, USA) and detected using a BioSpectrum® imaging system (UVP).

### 4.5. Small Interfering RNA (siRNA) Transfection

The p38 MAPK siRNAs were purchased from Ambion (Life Technologies, Carlsbad, CA, USA). MDA-MB-231 cells in 6-well plates (1 × 10^5^ cells/well) were cultured overnight and then treated with a mixture of 25 pmole siRNAs using Lipofectamine™ RNAiMAX transfection reagent (Life Technologies, Carlsbad, CA, USA) according to the manufacturer’s recommendations. The mixture contained silencer select pre-designed (ID s3585) and validated siRNAs (ID s11156) against human alpha p38 (MAPK 14) and beta p38 MAPK (MAPK11) to knockdown p38 MAPK expression. After 6 h, the cell supernatants containing the transfection reagents were replaced with fresh cell medium, and the cells were further incubated overnight. Nontargeting siRNA was also used as a transfection control. The solvent (DMSO) or 15 μM flavopereirine was separately added to MDA-MB-231 cells that were transfected with p38 MAPK or nontargeting siRNA. After incubation for 48 h, the cells were collected and harvested for Western blotting.

### 4.6. Plasmid DNA Transfection

The wells of 6-well plates were seeded with 2 × 10^5^ cells and further incubated overnight, and then the cells were transfected with DNA-lipid complexes using Lipofectamine™ 3000 and P3000™ transfection reagents (Life Technologies, Carlsbad, CA, USA) with modified protocols based on the manufacturer’s guidelines [13]. Briefly, DNA-lipid complexes were prepared in two tubes that contained 125 μL of Opti-MEM™ medium. In tube one, Opti-MEM™ medium was mixed with the Lipofectamine™ 3000 reagent (4.5 μL). In tube two, plasmid DNA (1.5 μg) was diluted with Opti-MEM™ medium and then mixed with P3000™ reagent (3 μL). Before being added to the cells, the two tubes were mixed and incubated for 10 min. The pHRIG-Akt1 plasmid (pAKT, a construct with a constitutively active human Akt1) was purchased from Addgene (Watertown, MA, USA) and pBSSK was an empty vector used as a negative control. The medium changes and flavopereirine treatment were performed according to the procedures for p38 MAPK siRNA transfection. Then, the cells were collected and harvested for Western blotting.

## Figures and Tables

**Figure 1 ijms-21-05362-f001:**
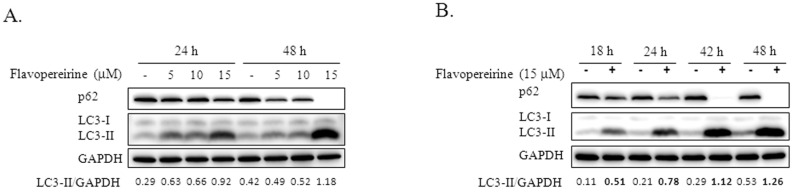
Flavopereirine induced LC3-II accumulation in dose- and time-dependent manners. (**A**) MDA-MB-231 cells were treated with various concentrations of flavopereirine (5-15 μM) for 24 and 48 h. (**B**) MDA-MB-231 cells were incubated with 15 μM flavopereirine for 18, 24, 42, and 48 h. LC3-II accumulation and p62 expression were evaluated by Western blotting. glyceraldehyde-3-phosphate dehydrogenase (GAPDH) was used as the internal control. The data are representative of three independent experiments. The band signals were quantified and the ratio of LC3-II related to GAPDH is presented under the blots.

**Figure 2 ijms-21-05362-f002:**
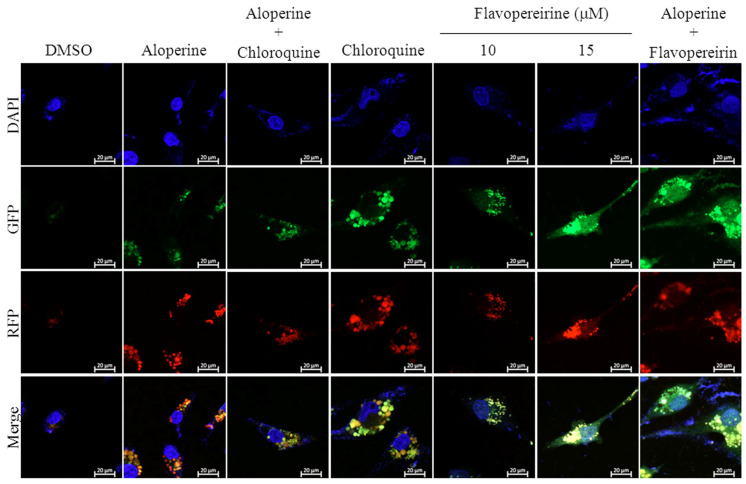
Flavopereirine inhibited autophagic flux in MDA-MB-231 cells. Representative fluorescent images of autophagosomes and autophagolysosomes in MDA-MB-231 cells were transduced with tandem RFP-GFP-LC3B and then independently treated with aloperine (100 μM), chloroquine (25 μM), flavopereirine, or a combination of the two drugs. In the merged images, autophagosomes are presented as yellow or orange puncta (RFP-GFP-LC3B), whereas red puncta (RFP-LC3B) indicate autophagolysosomes because acidification abolishes the green fluorescence. Images were obtained using a 63× oil immersion objective.

**Figure 3 ijms-21-05362-f003:**
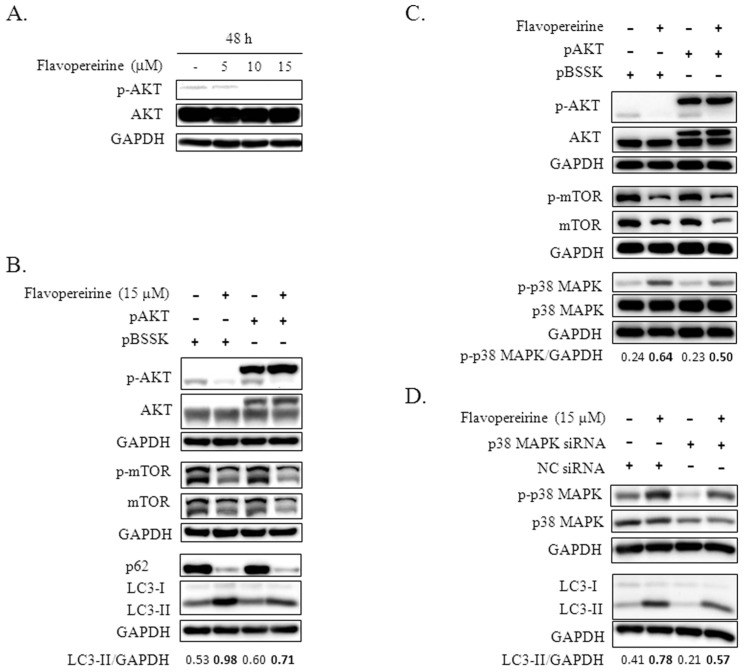
Flavopereirine inhibited autophagy through AKT/p38 MAPK signaling. (**A**) The expression level of p-AKT in MDA-MB-231 cells treated with increasing concentrations of flavopereirine was detected by Western blotting. (**B**, **C**) MDA-MB-231 cells transfected with plasmid constitutively expressing p-AKT (pAKT) or control plasmid (pBSSK) were treated with flavopereirine for 48 h and then analyzed for the expression of the indicated protein and their phosphorylation levels by Western blotting. (**D**) MDA-MB-231 cells transfected with p38 MAPK or negative control (NC) siRNAs were further treated with flavopereirine for 48 h. Western blotting was performed to detect p-p38 MAPK, p38 MAPK knockdown, and LC3 levels. The band signal was quantified and the ratio of LC3-II or p-p38 MAPK related to GAPDH is presented under the blots.

**Figure 4 ijms-21-05362-f004:**
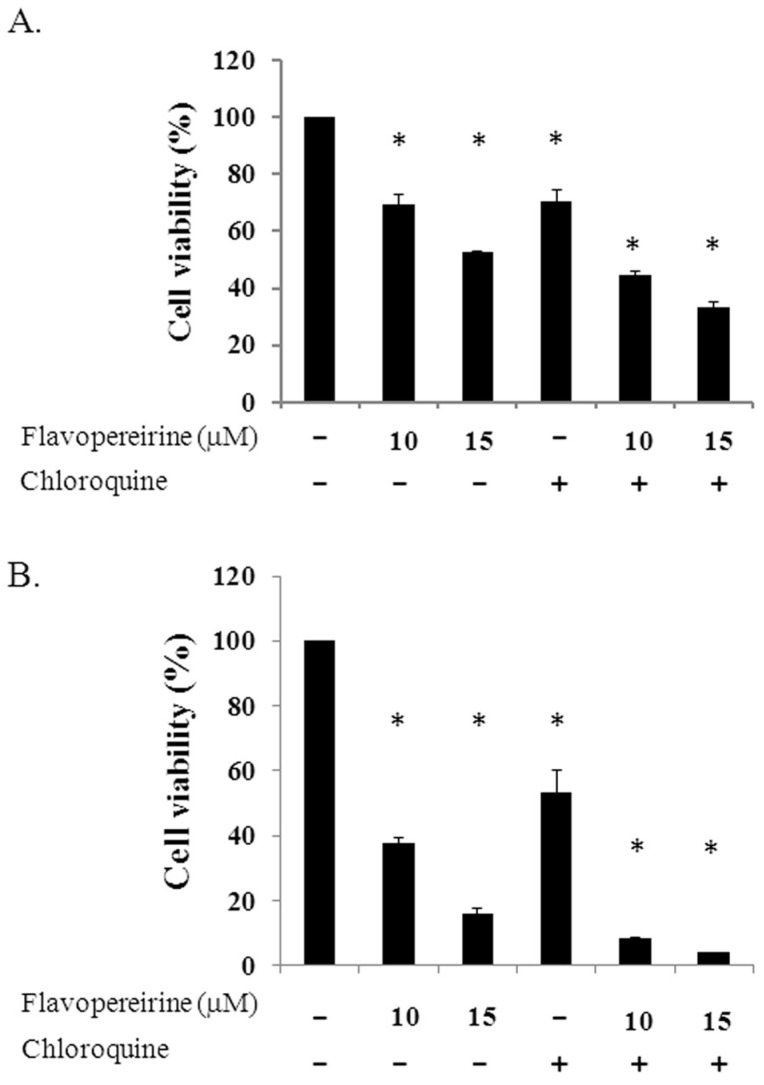
Cell viability of MDA-MB-231 cells treated with individual or a combination of flavopereirine with chloroquine. Cells (4 × 10^5^/well) were treated with different concentrations of flavopereirine, chloroquine (25 μM), or a combination of the two drugs, and cell viability was evaluated by a cell counting kit-8 (CCK-8) after treatment for 24 h (**A**) and 48 h (**B**). Histograms show the mean ± SD of three independent experiments performed in triplicate. * Indicates a significant difference from its respective untreated control as analyzed by one-way ANOVA followed by a Bonferroni test (*p* < 0.05).

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
