# Peer review of "Flavopereirine Inhibits Autophagy via the AKT/p38 MAPK Signaling Pathway in MDA-MB-231 Cells"

_ijms, 2020, doi:10.3390/ijms21155362_

Round 1

Reviewer 1 Report

This manuscript by Chen et al investigates the impact of flavopereirine on the process of autophagy in MDA-MB-231 triple negative breast cancer cells. This manuscript builds on their previous work which analyzed the effects of flavopereirine on cell cycle arrest and apoptosis.  There is a growing amount of interest in targeting autophagy for the treatment of cancer.  This manuscript is well-written, requiring only a few minor grammar edits, but the following comments are suggestions for improvement:

  1. The western blots in figure 1 are blurry and the text is difficult to read.
  2. In Figure 2, showing only 1-2 cells per condition is not enough to convince the reader of the authors’ findings. The data should be supported by quantification of fluorescence signal across multiple biological replicates, either through counting dual-positive cells or through flow cytometry.
  3. The westerns blots in Figure 3 are not as convincing as the text would have you believe. In many instances, differences (such as LC3-II levels in panel C) are unconvincing or even non-existent. This is further complicated by the data in Figure 4, which shows that up to 90% of the cells at the drug concentration and time point used in Figure 3 are actually apoptotic or non-viable. Thus, the authors seem to be doing a western blot on dead cells. This entire figure should be repeated at an earlier timepoint to avoid the complication of dead cells. Furthermore, the western blots should be quantified by densitometry and repeated across multiple biological replicates for statistical analysis.
  4. The authors use a p-Akt over-expression plasmid that does not seem to be mentioned in the methods. In particular, this plasmid seems to use a tagged version of the protein, based on the band-shift in the westerns in Figure 3. In the methods, the authors describe how they did the transfection, but they do not describe the DNA plasmid constructs they used.

Author Response

  1. The western blots in figure 1 are blurry and the text is difficult to read.

Ans: Thank you for this comment. We modified the following sentences in the Results and Figure 1:

Lines 81-82 in the Results: The autophagic flux is often examined based on the turnover of the autophagy marker LC3 (LC3-II) in western blot analyses. We tested the significance of flavopereirine in autophagy in MDA-MB-231 cells…

Lines 97 and 109-110 in the legend of Figure 1: Flavopereirine induced LC3-II accumulation in a dose- and time-dependent manner…The band signals were quantified, and the ratio of LC3-II relative to GAPDH is presented under the blots.

  1. In Figure 2, showing only 1-2 cells per condition is not enough to convince the reader of the authors’ findings. The data should be supported by quantification of fluorescence signal across multiple biological replicates, either through counting dual-positive cells or through flow cytometry.

Ans: Thank you for this comment. We actually performed multiple biological replicates and found that almost all cells treated with aloperine were red-positive in different objective fields. Moreover, the cells treated with flavopereirine and chloroquine showed a consistent dual-positive pattern as shown in the picture below. The images were obtained using a 20× objective. Flow cytometry may not be easy to perform due to the large proportion of cell death.

Fig. 1. Flavopereirine inhibits autophagic flux in MDA-MB-231 cells. Representative fluorescent images of autophagosomes and autophagolysosomes in MDA-MB-231 cells transduced with tandem RFP-GFP-LC3B and then independently treated with aloperine (100 mM), chloroquine (CQ, 15 mM), or flavopereirine. In the merged images, the autophagosomes are presented as yellow or orange puncta (RFP-GFP-LC3B), whereas the red puncta (RFP-LC3B) indicate autophagolysosomes because acidification abolishes the green fluorescence.

  1. The westerns blots in Figure 3 are not as convincing as the text would have you believe. In many instances, differences (such as LC3-II levels in panel C) are unconvincing or even non-existent. This is further complicated by the data in Figure 4, which shows that up to 90% of the cells at the drug concentration and time point used in Figure 3 are actually apoptotic or non-viable. Thus, the authors seem to be doing a western blot on dead cells. This entire figure should be repeated at an earlier timepoint to avoid the complication of dead cells. Furthermore, the western blots should be quantified by densitometry and repeated across multiple biological replicates for statistical analysis.

Ans: Thank you for this comment. We calculated the ratio of the key proteins relative to GAPDH in each panel, such as LC3-II/GAPDH (Figure 3). The upstream regulation of AKT targeting p-p38 MAPK was also demonstrated in our previous study as shown below (Eur J Pharmacol, 2019 Nov 15;863:172658). 

Fig. 2 MDA-MB-231 cells were transfected with a plasmid constitutively expressing p-AKT (p-AKT) or control plasmid (pBSSK) and then treated with flavopereirine for 48 h. The expression of p-AKT, AKT, p-p38 MAPK, and p38 MAPK was examined by western blot analyses. * indicates a significant difference (P < 0.05) between MDA-MB-231 cells transfected with pAKT or pBSSK and treated with flavopereirine as analyzed by a one-way ANOVA, followed by a Bonferroni test.

Although most of the population of cells treated with flavopereirine died (48 h), the cells included in the western blot analysis were collected after centrifugation. Therefore, only the intact cells can be collected in the cell pellet, and we increased the number of wells treated with flavopereirine to collect enough cells to perform the western blot analysis.

  1. The authors use a p-Akt over-expression plasmid that does not seem to be mentioned in the methods. In particular, this plasmid seems to use a tagged version of the protein, based on the band-shift in the westerns in Figure 3. In the methods, the authors describe how they did the transfection, but they do not describe the DNA plasmid constructs they used.

Ans: According to this suggestion, we added information regarding the p-AKT overexpression plasmid in the Materials and Methods.

Lines 303-305 in the Materials and Methods: The pHRIG-Akt1 plasmid (pAKT, a construct with constitutively active human Akt1) was purchased from Addgene, and pBSSK was an empty vector used as a negative control.

Reviewer 2 Report

The authors examined the effects of Flavopereirine as an autophagy inhibitor on human triple negative breast cancer MDA-MB-231 cells. They show that Flavopereirine inhibits autophagy flux via inhibition of nonclassical AKT/p38 MAPK signaling. The results are reasonable, whereas several points should be clarified.

Specific comments:

1) There is no comment on p62 even though they examined the expression.

2) in Figure 4B. Why was the percentage of cell viability at the right side minus?

3) Although the authors discussed on autophagic apoptosis/cell death, the cell death and cell arrest must be discriminated in examining the cell viability (Figure 4).

Author Response

The authors examined the effects of Flavopereirine as an autophagy inhibitor on human triple negative breast cancer MDA-MB-231 cells. They show that Flavopereirine inhibits autophagy flux via inhibition of nonclassical AKT/p38 MAPK signaling. The results are reasonable, whereas several points should be clarified.

Specific comments:

  • There is no comment on p62 even though they examined the expression.

Ans: According to this suggestion, we added a related description in the Discussion.

Lines 181-186 in the Discussion: In contrast to LC3-II accumulation, flavopereirine inhibited p62 in a dose-dependent manner (Figure 1). Previously, p62 was considered an autophagy indicator due to its ability to deliver polyubiquitinated proteins for autophagy degradation. Recent reports have demonstrated that p62 participates in the regulation of diverse cellular signaling pathways [17], which may interfere with p62 expression or degradation independent of autophagy. These findings suggest that the p62 level alone is not a good indicator of autophagy.

  • in Figure 4B. Why was the percentage of cell viability at the right side minus?

Ans: Thank you for this comment. We repeated the cell viability (48 h) in Figure 4B and obtained new results as shown below.

3) Although the authors discussed on autophagic apoptosis/cell death, the cell death and cell arrest must be discriminated in examining the cell viability (Figure 4).

Ans: Thank you for this comment. The detection of cell viability could include the cell population in cell cycle arrest. Our previous study actually found that there was significant accumulation of MDAMB-231 cells in the S phase at 48 h after the treatment with flavopereirine. We added the following description in the Discussion:

Lines 222-224 and 229 in the Discussion: Moreover, flavopereirine can lead to a significant accumulation of S phase (from 23.5% to 52.9%) at the same time point [13]…These data suggest that the decrease in cell survival was due to flavopereirine-induced apoptosis, S phase arrest and autophagy-related cell death.

We also used a Lactate Dehydrogenase (LDH) Cytotoxicity Detection KitPLUS (Roche) to determine flavopereirine-induced cell death according to the manufacturer’s instructions. However, the results were much lower than those observed in the assessment of flavopereirine-induced cell death because we found that most cells in the cell population lost their morphology, showed cell blebbing or detached from the bottom of the culture plates. One reason is that the half-life of LDH at 37°C is approximately 9 hours, but we investigated the effect of favopereirine for 48 h. The favopereirine-induced cell death observed in our previous study was also included in the Discussion.

Lines 218-221 in the Discussion: In our previous study, we used a pancaspase inhibitor (z-VAD-fmk) to pretreat MDA-MB-231 cells, which significantly abolished flavopereirine-induced apoptosis from 77.2 to 27.5% at 48 h [13]. Furthermore, the proportion of cells that were positive for PI but not Annexin-V staining was 3.8%.

Round 2

Reviewer 1 Report

The authors have adequately addressed this reviewer's concerns.